# Novel Intumescent Flame Retardant Masterbatch Prepared through Different Processes and Its Application in EPDM/PP Thermoplastic Elastomer: Thermal Stability, Flame Retardancy, and Mechanical Properties

**DOI:** 10.3390/polym11010050

**Published:** 2018-12-31

**Authors:** Weidi He, Ying Zhou, Xiaolang Chen, Jianbing Guo, Dengfeng Zhou, Shaopeng Chen, Meng Wang, Lingtong Li

**Affiliations:** 1National Engineering Research Center for Compounding and Modification of Polymer Materials, Guiyang 550014, China; hwd3301932@163.com (W.H.); zhouying_0304@126.com (Y.Z.); wm910625@163.com (M.W.); 2Key Laboratory of Advanced Materials Technology Ministry of Education, School of Materials Science and Engineering, Southwest Jiaotong University, Chengdu 610031, China; 13452713585@163.com (S.C.); lingtongli1994@163.com (L.L.); 32011 Special Functional Materials Collaborative Innovation Center of Guizhou Province, Guizhou Institute of Technology, Guiyang 550003, China; zhoudengfeng@git.edu.cn; 4Key Laboratory of Light Metal Materials Processing Technology of Guizhou Province, Guizhou Institute of Technology, Guiyang 550003, China

**Keywords:** ethylene-propylene-diene monomer, polypropylene, reactive extruding, intumescent flame retardant

## Abstract

In this work, the ethylene-propylene-diene monomer/polypropylene (EPDM/PP) thermoplastic elastomer filled with intumescent flame retardants (IFR) is fabricated by melting blend. The IFR are constituted with melamine phosphate-pentaerythritol (MP/PER) by compounding and reactive extruding, respectively. The effects of two kinds of MP/PER with different contents on the thermal stability, flame retardancy, and mechanical properties of materials are investigated by Fourier transform infrared (FTIR) spectroscopy, thermogravimetric analysis (TGA), limiting oxygen index (LOI), UL-94, cone calorimeter test (CCT), and scanning electron microscopy (SEM). FTIR results show that the reactive extruded MP/PER partly generates melamine pyrophosphate (MPP) compared with compound masterbatches. TGA data indicate that the best thermal stability is achieved when the molar ratio of MP/PER reaches 1.8. All the reactive samples show a higher flame retardancy than compound ones. The CCT results also exhibit the same trend as above in heat release and smoke production rate. The EPDM/PP composites filled with 30 and 35% reactive MP/PER exhibit the improved flame retardancy but become stiffer and more brittle. SEM photos display that better dispersion and smaller particle size are obtained for reactive samples.

## 1. Introduction

Thermoplastic elastomer (TPE) is a kind of material combining high elasticity and toughness, and can be processed by the method of thermoplastic base resin [1,2]. Compared with traditional rubber materials, the preparing process of TPE has a low energy dissipation and high efficiency, and can be high effectively reclaimed. Therefore, TPE has been widely used in automobiles, construction, insulators, electric appliances and so on [3,4,5,6]. As one kind of TPE, the ethylene–propylene–diene monomer/polypropylene (EPDM/PP) thermoplastic elastomer is usually prepared by dynamic vulcanization. EPDM is copolymerized by ethylene, propylene and unconjugated diene in which the saturated ethylene and propylene constitutional units in EPDM lead to a nice weather fastness, heat resistance and ozone resistance [7,8,9]. One of the two double bonds in the unconjugated diene can play as active center to conduct a cross-linking reaction [10]. In addition, as a plastic component, PP has a relatively high mechanical properties, good resistance of heat, insulativity and fatigue. Meanwhile, it is also cheap and easy to process [11].

Higher demand has been proposed in the properties of EPDM/PP to satisfy the increasing application requirements, especially for flame retardant fields. EPDM is a kind of typical combustible material and its limited oxygen index (LOI) value is only about 18%. Besides, there is a large difference in the degradation temperature between EPDM and PP, therefore, it is very difficult to find a suitable flame-retardant system for both of them. In order to improve the flame retardancy of polymers, polymer composites filled with flame-retardants have been known as an effective method. The usual halogen flame retardants have excellent flame-retardant performances, but the produced corrosive gas and a large amount of smoke limit their applications in many fields [12,13]. Metal hydroxides also have good flame retardancy, however, a shortcoming with the large loading proportion defects the mechanical properties of materials [14,15]. Recently, intumescent flame retardants (IFR) have been considered to be a development direction of flame retardant fields and a kind of environmental friendly flame retardant additive with the advantages of low smoke, halogen-free, no melt dripping, and excellent flame retardant efficiency [16,17,18]. The IFR system is usually consisted with nitrogen or phosphorus, which can form uniform and dense char or foam layers to isolate heat and oxygen as condensed phase in the combustion process. The low additive amount and high efficiency make them be a favorable development prospect in the flame retardant PP composites [19]. Among these IFR systems, the pentaerythritol and melamine phosphate (MP/PER) have been reported as a type of excellent halogen free flame retardant system, of which the effects on polymer materials are well investigated [20,21].

Although many investigations have been carried out on flame-retardant EPDM/PP composites [22,23,24], it is still necessary to study the flame retardancy further on in the consideration their mechanical properties, especially for the application of IFR system on the thermoplastic elastomer. In this work, MP/PER as a halogen free flame retardant is used to constitute an IFR system by compounding and reactive extruding, respectively. The thermal stability, flame retardancy and mechanical properties of EPDM/PP/IFR composites with the different molar ratio of PER and MP, processing method, and contents of IFR are investigated in detail by Fourier transform infrared (FTIR) spectroscopy, thermogravimetric analysis (TGA), limiting oxygen index (LOI), UL-94, and cone calorimeter test (CCT), scanning electron microscopy (SEM), and mechanical property tests. The main purpose of this work is to evaluate the effect of MP/PER IFR system on the EPDM/PP thermoplastic elastomer and seek to improve the flame retardancy and mechanical properties.

## 2. Experimental

### 2.1. Materials

Ethylene-propylene-diene monomer (EPDM, 4770P) was purchased from the Dow chemistry, US. Polypropylene (PP, T30s) was supplied by Sinopec Co, Ltd., Beijing, China. Pentaerythritol (PER) was purchased from Fuhua Chemical Materials Co., Ltd., Zhengzhou, China. Melamine phosphate (MP) was from Fine Collection Institute of Chemical Industry, Hefei, China. Dicumyl peroxide (DCP, Perkadox 14s-fl) was supplied by Akzo Nobel Chemicals Co., Ltd., Guangzhou, China; Triallyl isocyanurate (TAIC) was supplied by Shanghai Farida Chemical Co., Ltd., Shanghai, China. The naphthenic oil (KN4010) was provided by Shanghai Xiangping Industrial Co., Ltd., Shanghai, China.

### 2.2. Preparation of Flame Retardant Masterbatches

#### 2.2.1. Fabrication of Reactive Extruded Flame Retardant Masterbatches

The reactive masterbatches with 70 wt % MP/PER and 30 wt % PP were extruded on a two-screw extruder (TSE–40A, L/D = 40, D = 40 mm, Coperion Keya Machinery, Co., Ltd., Nanjing, China) at 260–270 °C. In these masterbatches, the molar ratio of MP/PER is 1.0, 1.2, 1.4, 1.6, 1.8, and 2.0, respectively.

#### 2.2.2. Fabrication of Compound Flame Retardant Masterbatches

The compound masterbatches with 70 wt % MP/PER and 30 wt % PP were fabricated on the same two-screw extruder above at 185–200 °C. In these masterbatches, the molar ratio of MP/PER is 1.8.

### 2.3. Preparation of the Flame Retardant Composites

Firstly, EPDM was soaked in naphthenic oil (30 wt % of EPDM) at room temperature for 24 h to improve the processability. The flashing point is usually applied to evaluate the volatility of naphthenic oil. The flashing point of the naphthenic oil in this work is 160 °C, and nearly most naphthenic oil can be volatilized during the processing of PP. Secondly, PP and EPDM were dried at 70 °C for 8 h. Then EPDM/PP masterbatches with cross-linking agent (DCP with 1.2 wt % of EPDM) and assistant crosslinker (TAIC with 1.5 wt % of EPDM) were extruded on the same two-screw extruder above at 185–200 °C. Finally, EPDM/PP thermoplastic vulcanizate and two kinds of MP/PER/PP flame retardant masterbatches were mixed and injection molded on an injection molding machine (CJ80MZ-NCII, Chengde Plastics Machinery Co., Ltd., China) at 195–210 °C. In order to obtain expected flame retardant and mechanical properties, the ratio of EPDM/PP in the final products was controlled at 40:60, which was adjusted by incorporation of PP.

### 2.4. Measurements and Characterization

#### 2.4.1. Fourier transform infrared spectroscopy (FTIR)

FTIR analysis was proceed on a NEXUS 570 type, Thermo Fisher Scientific Co., Ltd. (Waltham, MA, USA). The sample powders were tableted with KBr as the mass ratio of 1/100 before testing. Wavenumbers of the test were ranged from 400 to 4000 cm^−1^.

#### 2.4.2. Calculations of Conversion Rate

The reactive extruded MP/PER flame retardant masterbatches with different molar ratio were weighed as *W*_0_ by scale (2 ± 0.5 g). The samples were broken into small granulates and covered by filter paper and copper grid. Then they were set in a Soxhlet extractor (ethyl alcohol as extraction solution) to heat reflux for 8 h and cleanout by hot deionized water repeatedly. The dried extractives were weighed as *W*_l_. The conversion rate can be calculated as following equations:*W*_f_ = λ*W*_0_·*M*_0_/*M*_1_(1)
*C* = [1 − (*W*_0_ − *W*_1_)/*W*_f_] × 100%(2)
where *C* is the conversion rate; *W*_f_ is the initial weight of PER and MP; *M*_0_ is the total weight of PP, MP, PER before reaction; *M*_1_ is the total weight of reactive extruded products; λ is the weight percent of PER and MP before extruding, which is 70%.

#### 2.4.3. Scanning Electronic Microscopy (SEM)

SEM images of fracture surface were obtained by a Quanta FEG230 scanning electron microscope (FEI Co., Ltd., Hillsboro, OR, USA). SEM graphs of the composites were recorded after gold coating surface treating, with the accelerating voltage of 10 kV.

#### 2.4.4. Thermogravimetric Analysis (TGA)

The thermal properties of the composites were analyzed by thermogravimetric analysis (TGA) (Q-50 instruments, TA Co., Ltd., New Castle, PA, USA) under the gas flow of high purity grade nitrogen. About 5–10 mg of each sample was put in a platinum pan and heated from room temperature to 700 °C at a heating rate of 10 °C/min.

#### 2.4.5. Limiting Oxygen Index (LOI)

The LOI value was obtained by using an LOI instrument (type JF-3, Jiangning Analysis Instrument Factory, Nanjing, China) on sheets 120 mm × 6.5 mm × 3 mm according to the standard oxygen index test ASTM D2863-77. The LOI value is calculated according:LOI = (O_2_)/((O_2_) + (N_2_)) × 100%(3)

In which (O_2_) and (N_2_) are the concentration of O_2_ and N_2_, respectively.

#### 2.4.6. UL-94 Vertical Burning Test

The UL-94 vertical burning test was carried out by a CTF-2 vertical burning instrument (made in Jiangning, China) on sheets 127 mm × 12.7 mm × 3 mm according to ASTM D3801 standard. The three ratings (V-2, V-1, and V-0) were defined. The V-0 rating level stands for the highest requirement.

#### 2.4.7. Cone Calorimeter Test (CCT)

The cone calorimeter (Stanton Redcroft, UK) tests were carried out according to ISO-5660 standard procedures. Each specimen of dimensions 100 mm × 100 mm × 6 mm was wrapped in aluminum foil and exposed horizontally to an external heat flux of 50 kW/m^2^.

#### 2.4.8. Mechanical Properties

The tensile strength was measured by the material testing machine typed WDW-10C (Shanghai Hualong Test Instrument Co., Ltd., Shanghai, China) with the tensile speed of 50 mm/min. The bending strength was measured by the same machine, with the crosshead speed of 2 mm/min. All mechanical tests were performed at the room temperature of 23 ± 2 °C.

## 3. Results and Discussion

### 3.1. FTIR Analysis

Figure 1 shows FTIR spectra of PER, MP, compound and reactive PER/MP/PP flame retardant masterbatches. It is clearly seen from Figure 1A that several typical vibration peaks of PER are exhibited, such as the stretching vibration of –OH at 3314 cm^−1^, the stretching vibration of –CH_2_– at 2955 and 2886 cm^−1^, the bending vibration of CH_2_– at 1477 cm^−1^, and the stretching vibration of C–O at 1016 cm^−1^. For MP, the following vibration peaks are recognized, such as stretching vibration of –NH_2_ at 3393 cm^−1^, stretching vibration of –NH_3_^+^ at 3155 cm^−1^, stretching vibration of C–N at 1671 cm^−1^, and stretching vibration of P=O at 1108 cm^−1^. Meanwhile, the characteristic absorption peaks of (P=O)–OH are recognized at 2686 and 961 cm^−1^. After the reactive extruding, the FTIR spectrum of MP/PER/PP exhibits that the peaks of C–OH at 3369 and 1016 cm^−1^ of PER and (P=O)–OH at 2686 and 961 cm^−1^ disappear, and the two new peaks at 1089 and 1045 cm^−1^ appear, as shown in Figure 1D. This reflects the formation of the phosphoric ester groups (P=O)–O–C [23]. For the compound masterbatches, however, the two new peaks are not observed, and only a peak of P=O at 1108 cm^−1^ is recognized, which is attributed to MP. The reaction route between PER and MP in extruding can be speculated as shown in Scheme 1 and melamine pyrophosphate (MPP) has been partly generated.

FTIR spectra of reactive extruded MP/PER masterbatches with different molar ratios are investigated to find out the most effective ratio of MP/PER, and the results are also shown in Figure 2. The relative peak intensity of –NH_3_^+^ at 3167 cm^−1^ increases obviously when the MP/PER molar ratio increases from 1.0 to 2.0, which can be attributed to the –OH peak in PER at 3369 cm^−1^. In addition, the stretching vibration peak intensity of C–N at 1671 cm^−1^ also increases with the increasing of the molar ratio. It is also observed that the absorption peaks of (P=O)–O–C at 1089 and 1045 cm^−1^ increase with the increasing of the ratio until it reaches 1.8, however, it shows a decreasing trend when the ratio is higher than 1.8. This indicates that the molar ratio of MP/PER will clearly affect the conversion rate of the products and the highest extent of reaction is achieved when the molar ratio of MP/PER is 1.8.

The conversion rate of the reaction between MP and PER are calculated and the results are presented in Figure 3. It is also found that the conversion rate increases when the MP/PER molar ratio varies from 1.0 to 1.8, and then decreases. The highest conversion rate of 75.45% is obtained when the molar ratio reaches 1.8, which further confirms the FTIR results. The structures of the reaction products depend on the ratios of reactants. When the MP/PER molar ratio is greater than or equals 2.0, the hydroxyl groups of PER can be consumed completely and a product without hydroxyl groups will be produced. While the MP/PER molar ratio is less than or equals 1.0, the hydroxyl groups of PER cannot be consumed completely and a product with hydroxyl groups will be formed [21].

### 3.2. Thermal Stability

TGA curves of reactive MP/PER-PP masterbatches with different molar ratios at a heating rate of 10 °C/min under nitrogen are illustrated in Figure 4, and the detailed data are shown in Table 1. *T*_5%_ is defined as the onset decomposition temperature of sample which means 5% weight loss, and *T*_max_ as the primary peak temperature of DTG which means the highest weight loss rate. The TGA curves shift to higher temperature as the MP/PER molar ratio increases from 1.0 to 1.8, however, a dramatic shift to much lower temperature is observed when the molar ratio reaches 2.0. Generally speaking, MP/PER acts as an IFR with the sources of acid, carbon and gas. In the heating process, the acid source promotes the carbon source to form stable char layers which become bulk with the gas generated by the gas source [25]. These char layers covered on the surface of matrix can isolate the heat and flame to improve the thermal stability of composites. When the MP/PER ratio is low, carbon source is more sufficient than acid and gas sources so that the amount of generated MPP is also low. While the MP/PER ratio is too high (2:1), the decomposition of the large amount of gas source produces massive gas which can reduce the stability of char layers. This indicates that the molar ratio of MP and PER is responsible for the productivity of MPP. With the regard to the issue including the analysis in the previous chapter, the MP/PER molar ratio is set to 1.8 in the IFR system for the following work.

TGA and DTG curves of reactive extruded EPDM/PP/IFR (MP/PER = 1.8) with different IFR contents are presented in Figure 5, and the corresponding TGA data are listed in Table 2. *T*_5%_ is defined as the onset decomposition temperature of sample which means 5% weight loss, and *T*_max_ as the primary peak temperature of DTG which means the highest weight loss rate. It is found from Figure 5 that all the samples show two DTG peaks at about 300–500 °C. The first peak is much lower than the primary one and they contain the process including the volatilization of residual naphthenic oil (soaked in preparation), the decomposition of melamine phosphate and the formation of polyphosphoric acid [26]. The second peak at about 500 °C reflects the main degradation stage of EPDM/PP. The *T*_5%_ value of the composites with 35 wt % IFR increases obviously from 273.0 to 348.4 °C, and TGA curves shift to high temperature. However, the peak value of DTG decreases, and the value of *T*_max_ increases. This indicates that the increase of MP/PER IFR content promotes the formation of char layers to isolate heat flow and inhibit the volatilization of small molecule products, which improves the thermal stability of composites.

Figure 6 shows TGA and DTG curves of reactive extruded and compound EPDM/PP/IFR composites with 25 wt % content of IFR (MP/PER = 1.8). The thermal stability of the compound sample is lower than the reactive one at low temperature (250–350 °C), however, the curve of the reactive sample shifts to low temperature at high temperature (near 500 °C). The charred residue of the compound sample is also higher than that of reactive one. The values of *T*_5%_ and *T*_max_ of the compound sample are 302.6 and 493.0 °C, respectively. But the TGA results in high temperature mean that a better thermal stability is obtained for the compound sample. It is inferred from TGA that the gas flow of high purity grade nitrogen isolates the oxygen, which leads a different thermal degradation during the combustion process in atmosphere. The high char residues for compound samples are mainly caused by the carbonization of PER and polymer matrix. This will be further analyzed in the following chapter.

### 3.3. Flame Retardancy

The LOI and UL-94 levels of reactive and compound EPDM/PP/IFR composites are displayed in Table 3. It shows quite different flame retardancy between compound and reactive MP/PER flame retardant EPDM/PP. The LOI values of all the samples increase with the increasing of the content of IFR. At the same time, it is clearly seen from Table 3 that the samples with reactive MP/PER masterbatches have higher LOI values than compound ones. It can be found from Table 3 that all of the compound samples (content up to 35 wt %) fail in the UL-94 test, however, the samples with 30 and 35 wt % reactive MP/PER can reach V-1 and V-0 level, respectively.

The CCT is an advanced calorimetry instrument based on oxygen consumption theory in the field of fire science [27]. It is very effective to evaluate the combustion performances of materials, which provides an important reference in terms of material design, evaluation and fire prevention [28,29]. The curves of heat release rate (HRR), total heat release (THR), and smoke production rate (SPR) for EPDM/PP composites with the compound and reactive IFR are shown in Figure 7, and the detailed data are listed in Table 4. The materials will be at serious fire risk if the peak value of HRR is reasonably high [30]. In Figure 7A, a much lower and smooth HRR curve is observed in a reactive extruded sample, and the THR value is also lower than compound one as shown in Figure 7B. What’s more, the time to ignite (TTI) of reactive sample is longer than that of compound one, as listed in Table 4. In the SPR curves of Figure 7C, in the early stage of combustion (0–200 s), the reactive sample reaches a high smoke production, however, the SPR of the compound one becomes much higher than the reactive one in the following stage (200–700 s). The high smoke production rate is also related to the high fire risk.

The above results from UL-94 level, LOI and CCT indicate that the reactive extruded MP/PER shows higher flame retardant efficiency than the compound one in the flame retardant EPDM/PP composites. In reactive extruding, MP and PER partly generate MPP, in which the acid, carbon and gas sources gather on a same molecule. This makes it easy for carbon sources to form char layers under the catalysis of the acid source in heating process. These char layers can be huffed by the gas source to form a bulk and dense structure which can restrain the combustion of matrix and the volatilization of micromolecular products.

Figure 8 and Table 5 exhibit the effect of different contents of reactive MP/PER IFR on the dynamic flammability of EPDM/PP, and the photos of char residues after CCT are shown in Figure 9. For EPDM/PP, one single high peak of HRR is observed in Figure 8A. With the addition of MP/PER, two low peaks are emerged, and the peak values become much lower and smoother with the increasing content of IFR. The TTI values of the samples also increase when the IFR content is increased. In Figure 8B, the THR value of the sample with 20 wt % IFR is even higher than EPDM/PP after 600 s. Although the IFR can promote the formation of char layers which play a protective role for polymeric matrix, the heat release of “second burning” will still occur if the char layers are insufficient and dispersed [31]. The THR values exhibit a remarkable decrease when the IFR content reaches 30 wt %, and the SPR values also decrease obviously when the IFR content reaches 35 wt %, as shown in Figure 8C. The char layers are thick and dense enough to prevent the volatilization of micro-molecules. It is also concluded that the flame retardancy of EPDM/PP/IFR composites is tremendously improved when the contents of reactive MP/PER are over 30 wt %. This has been also verified by the results of UL-94 level and LOI values. In addition, the photos of charred residues in Figure 9B,D show that the composites with reactive MP/PER masterbatches have more stable and dense char layers, and the char residues in compound sample are fragile and splintered.

### 3.4. Mechanical Properties

The variation of mechanical properties is an important indicator for the applications of polymer composites. The tensile strength, bending strength, and elongation at break of EPDM/PP/IFR composites with different contents of reactive MP/PER IFR are shown in Figure 10. It can be seen from Figure 10 that the bending strength of the samples with higher contents of IFR increases significantly, which indicates that the stiffness and rigidity of composites are enhanced. The tensile strength shows an increasing trend and elongation at break decreases slightly when the IFR content is lower than 30 wt %. When it gets higher than 30 wt %, an obvious decrease in tensile strength and elongation at break is clearly observed. The variations in the mechanical parameters indicate the embrittlement has occurred. The effects of the inert IFR particles on mechanical properties are mainly attributed to the particle size and dispersion in the polymer matrix. When the content of IFR is high enough, the gathering of the particles makes them become hard spots. In loading process, these hard spots will cause the stress concentration, crack and lead to a failure of materials and mechanical performances deterioration [32].

The data of mechanical properties listed in Table 6 show that reactive sample has a higher tensile strength and elongation at break than compound one. This means that the reactive IFR can keep a higher retention rate of mechanical properties due to the lesser size of particles and better dispersibility after reactive extruding. The surface morphologies of the samples are exhibited in the SEM photos of Figure 11. Some gathered particles and pits sized from several micrometers to a dozen are observed in Figure 11A,B for the compound sample. For the reactive sample, although there are a few gathered particles, many more particles smaller than 1 μm can be observed during dispersion, as shown in Figure 11C,D. By the means of the reactive extruding at higher temperature, the particles of MP, PER and generated MPP can become smaller and disperse more uniformly to obtain high mechanical properties.

## 4. Conclusions

The effects of compound and reactive MP/PER intumescent flame retardant on the thermal stability, flame retardancy, and mechanical properties of EPDM/PP thermoplastic elastomer are investigated and discussed in detail. The FTIR results and the conversion rate show that the reactive extruding of MP/PER masterbatches partly generates MPP at high temperature. TGA results of flame retardant masterbatches show that the value of 1.8 is considered to be the most effective molar ratio of reactive MP/PER. All the UL-94 levels and LOI values of reactive extruded sample are higher than that of compound one. An addition of 35 wt % reactive MP/PER can greatly improve the flame retardancy of the composites. The CCT data also display the same trend as above in HRR, THR and SPR. For the generated MPP in the reactive extruding, the acid, carbon and gas sources are gathered on a same molecule, which makes it easy for the carbon source to form char layers under the catalysis of the acid source in the heating process. The increase of bending strength and decrease of elongation at break with the increasing of the IFR content indicate that the composites become stiffer and more brittle. The SEM photos of surfaces of samples display that a better dispersion and smaller particle size are obtained in reactive samples, which proves their higher retention rate of mechanical properties than found in compound ones.

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
