# Peer review of "Novel Intumescent Flame Retardant Masterbatch Prepared through Different Processes and Its Application in EPDM/PP Thermoplastic Elastomer: Thermal Stability, Flame Retardancy, and Mechanical Properties"

_polymers, 2018, doi:10.3390/polym11010050_

Round 1

Reviewer 1 Report

 This manuscript has been improved and thus it can be published in Polymers.

   However, I am not fully satisfied of all corrections.

   References 25 and 26 should be cited in Introduction, since this work is a simple continuation of quite old publications concerning fire retardancy studies of polypropylene (PP) IFR blends with melamine phosphate (MP) and pentaerythritol (PER).

   A content of a naphtenic oil may affect on char residues. An addition of a range of boiling point of naphtenic oil KN 4010 would be also valuable.

   In lines 174 and 175 a statement: "the absorption peaks of (P=O)-O-C at 1089 and 1045 cm-1 decrease with increasing the ratio until it reaches 1.8, ..." is not truth", because intensities of these peaks increase.

   What were FTIR spectra of MP/PER/PP composites extruded below at 185-200 ⁰C ? It would be interesting to compare them with the FTIR spectra of reactive extruded MP/PER/PP masterbatches.

   I am also not satisfied with a terminology of "compound flame retardant masterbatches".

Author Response

1. References 25 and 26 should be cited in Introduction, since this work is a simple continuation of quite old publications concerning fire retardancy studies of polypropylene (PP) IFR blends with melamine phosphate (MP) and pentaerythritol (PER).

Answer: Thank you for your good comments and suggestion. We have cited References 25 and 26 in this introduction according to your suggestion in this revision.

2. A content of a naphthenic oil may affect on char residues. An addition of a range of boiling point of naphthenic oil KN 4010 would be also valuable.

Answer: The 30wt% of naphthenic oil charged in EPDM is just used to improve the processability. The flashing point are usually applied to evaluate the volatility of naphthenic oil. The flashing point of the naphthenic oil in this work is 160oC, and nearly most naphthenic oil has been volatilized during the processing of PP. In addition, the content of a naphthenic oil is very small. So, we think that the charred residues of the composites are not affected by the content of naphthenic oil. This part has been added in experimental chapter in our revision.

3. In lines 174 and 175 a statement: "the absorption peaks of (P=O)-O-C at 1089 and 1045 cm-1 decrease with increasing the ratio until it reaches 1.8, ..." is not truth", because intensities of these peaks increase.

Answer: It is a writing error for this. Thanks for your findings. We have revised them in this revision.

4. What were FTIR spectra of MP/PER/PP composites extruded below at 185-200oC? It would be interesting to compare them with the FTIR spectra of reactive extruded MP/PER/PP masterbatches.

Answer: Thank you for your good suggestion. The mentioned spectra have been added in Figure 1 to show the comparison according to your suggestion in this revision.

5. I am also not satisfied with a terminology of "compound flame retardant masterbatches".

Answer: Thanks for your comments for this. We also thought about it carefully. The terminology of “compound” is only compared to “reactive” in this work. We have consult with a senior professor about it. Therefore, we still retained the terminology of “compound” in this manuscript. We would appreciate you if we can get your kind help about it.

Reviewer 2 Report

I would like to thank the authors for taking into consideration the suggested recommendations which could  improve the corresponding manuscript. However, the manuscript is still not suitable for publication in its current form and more experimental work is needed to support their results.

For example:

1) Figure 3: The authors did not show clearly, how they could measure the conversion rate. If they took the integration of any of the IR peaks, it will for sure does not give an accurate results as all peaks have interference with others.

2) The sentence of line 171-172 is not clear.

3) Reference 26 which is recently cited in the manuscript (please note that the page numbers of the cited reference are wrongly written and same mistake can be found in reference 25) reports the possible leaching of the target product which makes it feasible to use other analytical techniques such as NMR to follow the reaction.

4) Based on the cited articles, the effect of different molar ratios of MP/PER was already investigated and published. The authors in this manuscript repeated the same experiments and ended up wit the same conclusion. This will not add any additional value to the manuscript if the authors did not study for example the reason behind the decrease of the conversion if MP/PER ratio is 2 with comparison to others.

5) In tables 1, 2, 3 and 4: the error values should be mentioned in the tables. The values indicate one run experiments which can not be acceptable as representative values.

Author Response

1) Figure 3: The authors did not show clearly, how they could measure the conversion rate. If they took the integration of any of the IR peaks, it will for sure does not give an accurate result as all peaks have interference with others.

Answer: Thank you for your comments. The MP can be dissolved in ethyl alcohol and PER can be dissolved in water, however, the generated MPP cannot be dissolved in both of them. The Soxhlet extractor has been used to obtain the undissolved extractives to calculate the conversion rate as stated in experimental chapter. It is described in “Calculations of Conversion rate”.

2) The sentence of line 171-172 is not clear.

Answer: Thank you for your comments. We have rewritten this sentence according to your suggestion in this revision.

3) Reference 26 which is recently cited in the manuscript (please note that the page numbers of the cited reference are wrongly written and same mistake can be found in reference 25) reports the possible leaching of the target product which makes it feasible to use other analytical techniques such as NMR to follow the reaction.

Answer: Thank you for your good comments and suggestion. We have corrected these pages of References 25 and 26 in this revised manuscript.

4) Based on the cited articles, the effect of different molar ratios of MP/PER was already investigated and published. The authors in this manuscript repeated the same experiments and ended up wit the same conclusion. This will not add any additional value to the manuscript if the authors did not study for example the reason behind the decrease of the conversion if MP/PER ratio is 2 with comparison to others.

Answer: Thanks for your comments on this. PER/MP flame retardant is a kind of well investigated halogen free flame retardant system which runs very well in many thermal plastic polymers. In these cited articles, all the molar ratio including 1, 1.2, 1.4, 1.6, 1.8, and 2 of MP/PER filled in polymer matrix are investigated only by using UL-94 level and LOI. And the detailed analysis of FTIR, TGA and other properties are only proceeded on the MP/PER molar ratio of 1, 1.4, 1.6, and 2. It is lack of information on the masterbatches of more detailed molar ratios of MP/PER. What’s more, these articles are mainly focus on MP/PER IFR systems reinforced PP composites. However, in this work, the matrix are dynamic vulcanized EPDM/PP elastomers which makes a different in combustion behaviors. On the other hand, the carrier resin contents of masterbatches in the cited articles are relatively low (from 5-15%), and in this work this content are set as 30% which make the extruding process much more easily. A different conclusion of MP/PER molar ratio in EPDM/PP has also been made: the MP/PER molar ratio of 1.8 has been proved to be the best ratio for EPDM/PP matrix in this work, other than that of 1.6 for PP matrix in these cited articles. So, it is concluded that this work of this manuscript is different with previous publication.

5) In tables 1, 2, 3 and 4: the error values should be mentioned in the tables. The values indicate one run experiments which cannot be acceptable as representative values. 

Answer: Thanks for your good suggestion. The error values have been added in the average values of the Tables according to your suggestion in our revised manuscript.

Round 2

Reviewer 2 Report

I would like to thank the authors for considering all comments and suggestions. Accordingly, I recommend accepting the manuscript in its current form.